# Geographical Patterns in Functional Diversity of Chinese Terrestrial Vertebrates

**Xinyuan Sun** †, **Na Huang** † **and Weiwei Zhou** *

State Key Laboratory of Grassland Agro-Ecosystems, College of Ecology, Lanzhou University, Lanzhou 730000, China
* Correspondence: zhouweiwei@lzu.edu.cn
† These authors contributed equally to this work.

**Abstract:** Identifying priority regions is essential for effectively protecting biodiversity. China is one of the world's megabiodiversity countries, but its biodiversity is seriously threatened by anthropogenic forces. Many studies have identified priority regions in China for conserving biodiversity. However, most of these studies focused on plants and mainly relied on metrics such as species richness. A comprehensive assessment of functional diversity hotspots of Chinese terrestrial vertebrates is still lacking. In this study, we collected distribution information and functional traits of terrestrial Chinese vertebrates. We calculated functional richness and identified hotspots. Then, we assessed the overlap between functional hotspots and hotspots identified based on species richness. We found that the mountains in southern China harbor the most hotspots. Southwestern China is the most important region for biodiversity conservation, as it harbors functional diversity and species richness hotspots of multiple taxa. Mismatches between functional diversity and species richness hotspots were found in all taxa. Moreover, the locations of functional hotspots are different among taxa, even within taxonomic units. For example, the functional diversity patterns of Rodentia, Chiroptera and other mammalian taxa are different. These results highlight the importance of considering distinct groups separately in conservative actions.

**Keywords:** functional diversity; species richness; diversity hotspots; Chinese terrestrial vertebrates; conservation



## 1. Introduction

Biodiversity is being threatened on a global scale because of anthropogenic forces [1,2]. Due to human activities, such as unsustainable use of land, water and energy use; pollution and climate change, the extinction rate of species is approximately 1000 times higher than the background extinction rate [3,4]. China is famous for its rich biodiversity. As one of the world's megabiodiversity countries [5], China is home to 8201 vertebrate species, which includes 3232 terrestrial vertebrate species [6]. However, due to multiple reasons, a large proportion of these species are under threat. According to the latest assessment [7], the proportion of threatened species is approximately 26.4% for mammals, 10.6% for birds, 30% for reptiles, 43% for amphibians [8,9] and 20.4% for fishes [10]. The World Wildlife Fund identified the 200 most critical and endangered ecoregions, of which 46 are located in or intersect with China [11]. Among the 34 regions with high levels of biodiversity but are threatened by humans, four either intersect with or are in China [12,13].

How to effectively protect biodiversity is an important challenge for conservation biologists. To better conserve biodiversity with limited resources, it is essential to identify priority areas [14]. Many studies have identified priority regions in China. These studies identified diversity hotspots of different groups and investigated the potential impacts of environmental factors on biodiversity patterns [11,15–22]. These studies indicated that most hotspots are in mountainous regions, especially in the mountains of southern China. However, most studies have focused on plants. The comparison of hotspots among different

taxa finds extensive incongruences [22], which implies that the hotspots identified based on plants are insufficient to protect the biodiversity of animals. In addition, new species have been described recently, especially amphibian and reptile species [23]. It is necessary to update our knowledge about the spatial pattern of biodiversity [14]. It is also noticeable that most studies have identified hotspots based on species richness (SR), and few of them have identified hotspots using phylogenetic diversity [15]. Studies focusing on other facets of biodiversity, such as functional diversity (FD), are rare.

Biodiversity is a multifaceted concept [24]. Functional diversity, which represents the variety of functional traits within an assemblage, is a critical component of biodiversity [25]. It greatly affects ecosystem functioning, such as ecosystem stability and nutrient availability. As one of the most ecologically relevant measures, FD generally mirrors how species interact with other organisms and the environment [26]. Although phylogenetic diversity is assumed to be a good surrogate for FD [27,28], their relationships have also been questioned. The direct relationship between FD and phylogenetic diversity originates from the belief that evolutionary diversification produces trait diversification. However, the association between functional and phylogenetic diversities has been questioned. Weak spatial congruences between phylogenetic diversity and FD have been reported [29,30]. By analyzing the traits of vertebrate species, a previous study determined that phylogenetic diversity can be a very weak proxy for FD [31,32]. These studies further warned of the risk of using phylogenetic diversity in spatial conservation planning. Therefore, it is necessary to directly investigate FD based on functional traits.

Therefore, we reconstructed the FD and SR patterns of Chinese terrestrial vertebrates and identified hotspots. We aim to answer three questions. First, we tested whether functional hotspots and SR hotspots were congruent with each other. Second, as the ecological requirements and the functional traits differed among taxa, we calculated FD and SR for these groups separately and tested whether the hotspots were congruent among taxa. Third, it is common that one morphologically conserved clade constitutes a large proportion of species of a traditionally recognized taxonomic unit. For instance, the species number of Rodentia accounts for approximately 40% of mammals. We tested whether the diversity patterns of such a clade were congruent with those of other species.

## 2. Materials and Methods

### 2.1. Data Collection

We collected distribution data from different resources for each taxon. For mammals, we downloaded the distribution range maps from the International Union for Conservation of Nature (IUCN) Red List of Threatened Species [33]. For birds, the distribution data were obtained from BirdLife International (http://datazone.birdlife.org/species/requestdis, accessed on 14 December 2020) [34]. For amphibians, we first obtained the distribution range maps of species from the IUCN. As amounts of species have been described in recent years, we also collected information on species according to the species list from [23]. For species that were not recorded in the IUCN dataset, we collected distribution information from multiple resources, including original species descriptions, books, and online datasets (Table S1). For Squamata, we also collected information based on the species list from [23]. Range maps of species were extracted from the Global Assessment of Reptile Distributions (GARD) dataset [35]. The data from IUCN were downloaded in July 2019. We transformed the range maps and distribution points into a presence–absence matrix using the R package letsR v4.0 [36]. We projected the presence–absence matrix using Behrmann equal area projection, and the resolution of grid cells was set as $50 \times 50$ km$^2$. Then, we calculated the species richness of each grid. The grids with fewer than five species were removed from further analyses. However, for Caudata (Amphibia), the grids with three or more species were kept when calculating functional diversity, as the species richness was lower than five in most grids.

We collected functional traits for each taxon from different resources. For mammals, we used morphological, life history and ecological data from the most comprehensive

dataset focused on Chinese species [37]. For birds, we used a comprehensive dataset published recently, which covered eight traits, including body weight, body length, beak peak length, wing length, tail length, tarsus length, clutch number and egg volume [38]. For Squamata, we compiled data on snakes from previous studies [39–42], including body length, body mass and habitats. We also combined data from different resources, including books, species descriptions and online databases (Table S2). For other species, which are referred to as lizards here, we also used trait data from a recently published dataset [43]. For amphibians, we collected eight continuous measurements for Anura (Table S3), which included snout–vent length, head length, head width, eye diameter, the diameter of the tympanum, hand length, foot length and tibia length. For Caudata, we also collected eight traits, including total length, snout–vent length, head length, head width, eye diameter, tail length, forelimb length and hindlimb length (Table S4). Another 10 categorical traits (clutch size, egg size, breeding site, primary larval habitat, adult microhabitat, activity cycle, reproductive cycle, breeding time, parental care and fertilization type) were obtained from a recently published dataset [44]. In total, our data covered 1279 species of birds, 508 species of mammals, 404 species of Anura, 78 species of Caudata, 241 species of snakes and 194 species of lizards.

### 2.2. Diversity Metric Calculation and Hotspots Identification

We used functional richness (FRic), which is one of the most commonly used metrics, to represent FD [45–47]. FRic is a metric of the richness dimension that reflects the total differences among observations and was calculated by the functional space occupied by species. We first calculated the functional distance matrix using Gower's distance. Then we calculated FRic using the 'dbFD' function in the R package FD v1.0-12.1 [48], based on the species distribution data and the functional traits of the species. Body length was log-transformed, and other measurements were divided by body length before analyses. We identified FD hotspots using two methods. First, the top 5% of grids were selected as hotspots. Second, we tested whether the diversity of a grid was significantly high following the approach of [49]. We randomly selected 1000 grids as a random distribution. Then, we tested whether the functional richness of each grid was significantly higher than the random distribution. For Caudata, we selected 100 grids as random distribution as fewer grids had more than three species. We also identified diversity hotspots based on SR and compared them with hotspots identified based on FRic.

Based on their morphological and ecological differences, we divided the Chinese terrestrial vertebrates into six groups to perform the analyses: birds, mammals, Anura, Caudata, snakes and lizards. For amphibians, as the body plans were very different between Anura and Caudata, we calculated FRic separately for these two orders. Gymnophiona was not included in this study, as there is only one species in China, and its body plan was distinct from those of the other two orders [50]. For Squamata, as snakes are different from other species in body plan, ecological requirements and environmental drivers of biodiversity patterns [51], we also calculated FRic for snakes and lizards separately. For mammals, we first calculated Fric using all species. As Rodentia and Chiroptera comprised more than half of the species but occupied much less functional space, we also calculated the Fric of Rodentia and Chiroptera separately. Then, we reanalyzed the mammal data after removing the Rodentia and Chiroptera species. For Chiroptera, as the FD was greatly impacted by a few distinct species, we also reanalyzed the data after removing two species of the genus *Megaderma*. We used Schoener's D [52] and Hellinger's distance [53] to measure the similarity between Fric and SR. We calculated Schoener's D and Hellinger's distance following the approaches of [54]. We standardized the raster to make all values sum to one at first. Then, we calculated Schoener's D and Hellinger's distance between Fric and SR for each group.

## 3. Results

*3.1. Distribution Patterns of the Hotspots*

In general, the SR and FD patterns were similar (Figures S1 and S2). After removing the grids with too few species to calculate functional richness, we reconstructed spatial patterns of functional diversity based on 1279 species of birds, 508 species of mammals, 389 species of Anura, 47 species of Caudata, 236 species of snakes and 186 species of lizards. The mountainous regions in southern China had higher diversity. However, differences between SR and FD were also detected. For Chiroptera, the FD pattern was greatly impacted by two species of the genus Megaderma. After removing species of the genus Megaderma, southwestern China had a higher level of FD (Figure S3). For snakes, the southern Hengduan Mountains in Southwest China, Nanling Mountains, and Wuyi Mountains in Southeast China and downstream of the Changjiang River, harbored higher levels of diversity.

The two approaches we used to identify FD hotspots yielded similar results (Figures 1 and S4). It was clear that most functional hotspots were in the mountainous regions in southern China, especially in southwestern China (Figure 2). The FD hotspot locations were different among taxa (Figure 1). For birds, we identified three FD hotspots: the East Himalaya, Hengduan Mountains and the southwestern Yungui Plateau. For amphibians, the hotspots were different between Anura and Caudata (Figure 1). For Anura, we identified four hotspots, the southern Yungui Plateau, the northeastern edge of the Yungui Plateau, the Nanling Mountains and Hainan Island. For Caudata, Dabie Mountains in Central China harbored an FD hotspot. For Squamata, the hotspots were also different between snakes and lizards. For snakes, the hotspots were in the southern Hengduan Mountains, Nanling Mountains and mountains in Southeast China. For lizards, the southern Hengduan Mountains and Nanling Mountains were two large hotspots. Several smaller hotspots were also found in northern Taiwan Island, Hainan Island and East Himalaya. The functional traits were also different between these groups (Figure 3). For amphibians, the average body length of Caudata was larger than that of Anura, which was caused by their different body plans. For Squamata, the body lengths of snakes were much larger than those of lizards.

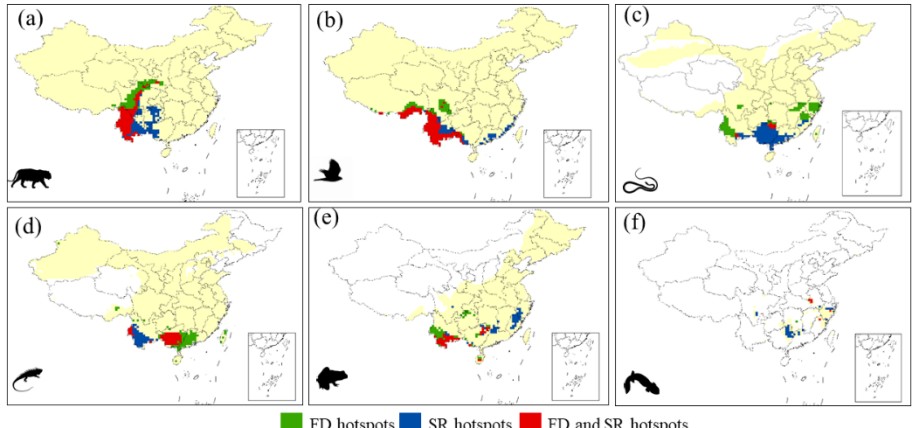

**Figure 1.** Geographic distributions of functional diversity (FD) hotspots and species richness (SR) hotspots. The green indicates FD hotspots, dark blue indicates SR hotspots and red indicates FD and SR hotspots. (**a**): Mammals; (**b**): Birds; (**c**): Snakes; (**d**): Lizards; (**e**): Anura; (**f**): Caudata. Silhouettes were downloaded from PhyloPic (www.phylopic.org, accessed on 14 December 2020). The map content approval number is GS(2019)1822.

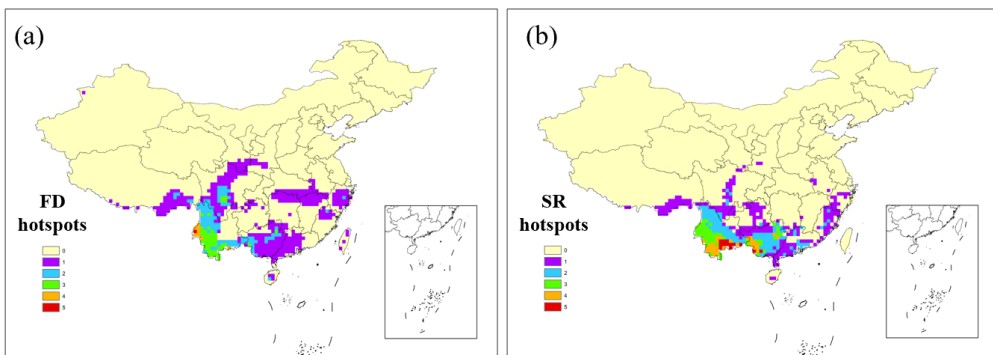

**Figure 2.** Overlapping of diversity hotspots among taxa. (**a**): Functional diversity hotspots; (**b**): Species richness hotspots. The map content approval number is GS(2019)1822.

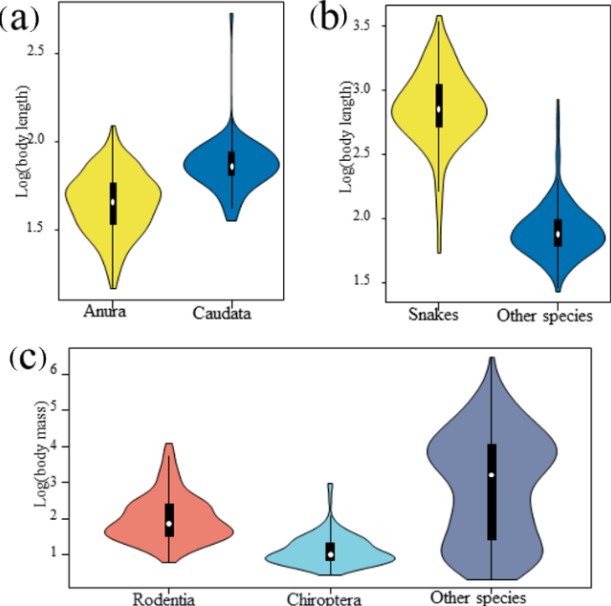

**Figure 3.** Differences of functional traits among taxa. (**a**): Body length of Anura and Caudata; (**b**): Body length of snakes and lizards. (**c**): Body mass of different groups of mammals.

For mammals, we identified one large FD hotspot (Figure 1), which was in southwestern China, east edge of The Qinghai–Tibet Plateau and Qinling Mountains. It was noticeable that the locations of functional hotspots of Rodentia and Chiroptera were different from the hotspots calculated based on all mammalian species (Figures 4 and S5). For Rodentia, two regions were identified as functional hotspots. For Chiroptera, as this pattern was caused by two species in the genus Megaderma (Figure S3), we analyzed the data after removing these species. After removing the two species, several functional hotspots were identified, which were in the East Himalayas. Several smaller hotspots were also found in South China and Hainan Island (Figure 4). Besides Rodentia and Chiroptera, the spatial pattern of functional hotspots of other mammalian species was similar to hotspots calculated based on all mammalian species (Figures 1 and 4). The functional traits, including body mass and diets, were different among Rodentia, Chiroptera and other species of mammals (Figures 3 and 5). The range of body mass was smaller in Rodentia and Chiroptera than in other species. The active time and diets were also different, especially for Chiroptera, which were all nocturnal species mainly eating invertebrates.

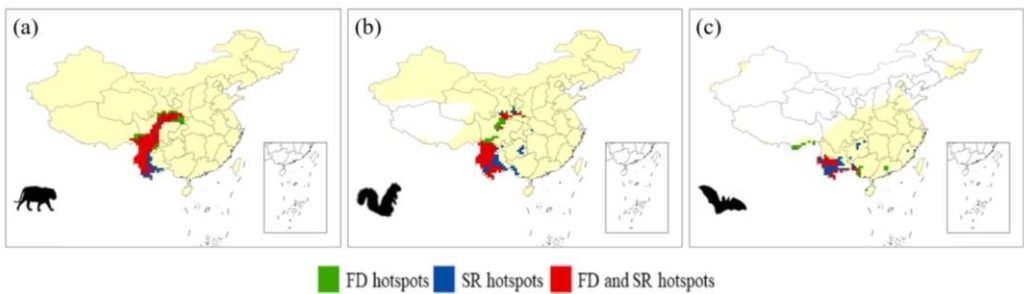

**Figure 4.** Geographic distributions of functional diversity (FD) hotspots and species richness (SR) hotspots of different groups of mammals. The green indicates FD hotspots, dark blue indicates SR hotspots and red indicates FD and SR hotspots. (**a**): Mammalian species besides Rodentia and Chiroptera; (**b**): Rodentia; (**c**): Chiroptera. Silhouettes were downloaded from PhyloPic (www.phylopic.org, accessed on 14 December 2020). The map content approval number is GS(2019)1822.

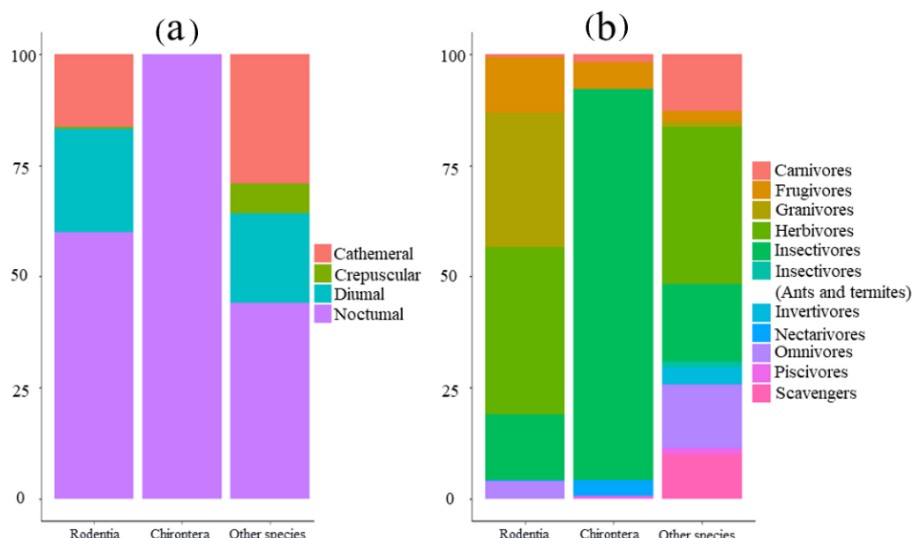

**Figure 5.** Differences of active time and diets among different groups of mammals. (**a**): Proportions of active time; (**b**): Proportions of diets.

### 3.2. Differences between FD and SR Hotspots

The general pattern of SR hotspots was similar to that of functional hotspots, and the mountainous regions in southern China harbored the most hotspots (Figure 2). However, for each group, the pattern was more or less different (Figure 1). We also calculated Schoener's D and Hellinger's distance between FRic and SR for each group (Table 1). For birds, SR and FD hotspots overlapped in Southwest China. However, regions near the northern Hengduan Mountains were functional hotspots but not SR hotspots. Meanwhile, the coastal regions in South China were SR hotspots but not FD hotspots. For amphibians, the hotspots of Anura identified by FRic and SR were similar in Southwest China. However, in Southeast China, the Wuyi Mountains are an SR hotspot but not an FD hotspot. For Caudata, a region of the Dabie Mountains in Central China was a hotspot for both FD and SR. In South China, Nanling Mountains harbored an SR hotspot, which was not an FD hotspot. For Squamata, the hotspots identified based on FRic and SR were also different. For snakes, FD hotspots were located in Southwest China and Southeast China, while SR hotspots were in the central part of South China, covering most regions of Guangxi Province and western parts of Guangdong Province. For lizards, Guangdong Province was an FD hotspot and Yunnan Province was an SR hotspot. Guangxi Province harbored both SR and FD hotspots. Meanwhile, several small FD hotspots were found in Hainan Island and northern Taiwan Island.

**Table 1.** Schoener's D and Hellinger's distance between functional diversity and species richness.

| Taxa | Schoener's D | Hellinger's Distance |
|---|---|---|
| Anura | 0.535 | 0.692 |
| Birds | 0.348 | 0.813 |
| Caudata | 0.606 | 0.46 |
| Lizards | 0.714 | 0.479 |
| Snakes | 0.731 | 0.425 |
| Mammals | 0.343 | 0.907 |
| Rodentia | 0.649 | 0.516 |
| Chiroptera | 0.804 | 0.274 |
| Mammalian speceis besides Rodentia and Chiroptera | 0.752 | 0.307 |

For mammals, the FD hotspot was much larger than the SR hotspot on the east edge of the Qinghai–Tibet Plateau. Meanwhile, the Qinling Mountains were identified as an FD hotspot but not an SR hotspot. In contrast, large areas in southern China were identified as SR hotspots but not FD hotspots. Notably, when placing mammalian species into three groups, SR and FD hotspots largely overlapped. For species besides Rodentia and Chiroptera, most SR and FD hotspots overlapped, except for some peripheral regions, such as the Qinling Mountains. For Rodentia, SR and FD hotspots overlapped in Southwest China. Two SR hotspots in the east were identified, which were not FD hotspots. Similar results were found for Chiroptera. FD hotspots were overlapping with SR hotspots in Southwest China. The East Himalayas and Nanling Mountains were FD hotspots but not SR hotspots. Schoener's D and Hellinger's distance obtained similar results. The FD and SR were more similar when the three groups were analyzed separately (Table 1).

## 4. Discussion

### 4.1. The Spatial Pattern of Functional Hotspots and Potential Causes

Most functional hotspots that we identified were in the mountains of southern China (Figures 1 and 2). Southwestern China is the most important region and is a common functional hotspot for different groups. This pattern is congruent with previous studies, although most of these studies focused on diversity hotspots of seed flora [11,16]. Several factors together make the mountains in southwestern China FD hotspots. First, the high level of heterogeneity in the topography and climate provides diverse habitats for different species. Southwestern China is famous for its complex topography [16,55]. The altitudes in this region range from nearly zero to more than 4000 [56]. The diverse climates include different climatic types ranging from tropical to temperate climates. Furthermore, different vegetation types exist in this region, ranging from tropical forest to alpine steppe [57], even semiarid savanna exists here [58]. For example, there are even eight vegetation types at different altitudes in one place in the Gaoligong Mountains [57]. The high level of environmental heterogeneity provides suitable habitats for different species, which may be one of the reasons for the higher FD in this region. Second, this region harbored several refuges during the Quaternary [59]. The stable climate and complex topography generated refuge for species during climatic changes [60]. The mountains in southwestern China form many sky islands, which are geographically isolated, high-elevation areas in continental mountain ranges. The sky islands in this region facilitate species dispersal along altitudes during climatic shifts, which prevents potential extinctions [56]. Then, the impacts of climatic changes are buffered, and relic species are preserved in this region. The refuge may also contribute to the formation of diversity hotspots. Third, the mix of species from different zoogeographic realms may be another reason for the formation of diversity hotspots [61]. Southwestern China is near the boundary between the Oriental and Sino–Japanese zoogeographic realms. The mix of species from different realms may allow highly divergent species to coexist together, which leads to a higher level of FD. In a previous study, the north–south dispersals of subtropical and temperate species were

considered a reason for the formation of biodiversity hotspots in southern China [15]. Previous studies found widespread dispersal in different taxa between southern China and Southeast Asia [62,63]. The faunal exchanges permit southwestern China to harbor species that are highly divergent. The coexistence of distinct species generates a large functional trait space and makes this region a functional hotspot.

### 4.2. The Differences between FD Hotspots and SR Hotspots

Species richness is a commonly used index in conservation. However, conservation plans based on SR alone are insufficient to protect biodiversity [25]. In this study, differences between hotspots identified based on SR and FD were detected in all taxa. We found that several mountains in the north were functional hotspots but not SR hotspots (Figure 2). For example, the Qinling Mountains were a functional hotspot for mammals but not an SR hotspot. Furthermore, the functional hotspots were larger than the SR hotspots on the eastern edge of the Qinghai–Tibet Plateau. For birds, we also found a functional hotspot near the eastern edge of the Qinghai–Tibet Plateau, which did not overlap with SR hotspots. For snakes, most FD hotspots were not overlapped with SR hotspots. The latitudinal diversity gradient is the most common diversity pattern. In our study, we also found that SR was higher in southern China (Figure S2). However, mountains in the north may have a higher level of FD. Although SR was lower in the north, the fauna covered all functional categories. For example, the Qinling Mountains were not an SR hotspot, but this region harbors diverse animals. The body mass of species in the Qinling Mountains covers almost the full range of Chinese mammals, from small mammals such as rodents to large herbivores such as *Budorcas taxicolor* [64]. Several unique species are also distributed here, such as pandas and snub-nosed monkeys [64].

### 4.3. Distinct Patterns within a Taxonomic Unit

It was noticeable that the FD patterns were different between groups within a taxonomic unit. Conservation strategies and plans are often made based on these taxonomic units, even for protecting FD [65]. However, conservation actions based on traditional taxonomic units may not be suitable to protect biodiversity, especially in regard to FD [25]. Our results highlight the importance of considering functionally distinct groups separately in conservative actions.

For Squamata, we found that the snake diversity pattern was different from that of lizards. Southwest China is an FD hotspot for snakes and SR hotspot for lizards. South China harbors FD and SR hotspots for lizards. In contrast, it is an SR hotspot but not an FD hotspot for snakes. Previous studies have indicated that both diversity patterns and environmental drivers of the patterns are different between snakes and lizards [51]. The body length, which is an important functional trait, is different between snakes and lizards (Figure 3). The diets are also different, as all snakes are carnivorous, while the diets of lizards are much more diverse [66]. Therefore, it is necessary to treat snakes and lizards as two groups to better protect their FD. The situation in Amphibia is similar. There are three orders with very distinct body plans. Our study compared the diversity patterns between Anura and Caudata. We found that the patterns of both FD and SR were very different between these two orders.

For mammals, the FD spatial patterns are different among Rodentia, Chiroptera and other species. It is noticeable that the FD patterns of all mammalian species mainly reflect the FD of taxa other than Rodentia and Chiroptera. Although Rodentia and Chiroptera constitute the largest proportion of mammal species, they occupy much less functional space than other species, which may be caused by their generalized and conserved body plan and feeding apparatus. Regarding body mass, which is one of the most important functional traits, the ranges of Rodentia and Chiroptera are within the range of other mammalian species (Figure 3). In regard to active time, almost all species of Chinese Chiroptera are nocturnal animals. The diets are also simpler in Rodentia and Chiroptera. Species of Rodentia feed largely on different parts of plants, and Chiroptera feed on invertebrates

(Figure 5). Therefore, conservation plans made based on the FD of all mammalian species may fail to protect the FD of Rodentia and Chiroptera. A previous study also indicated that Rodentia may be neglected in terms of conservation efforts [67]. Interestingly, the spatial patterns of FD and SR hotspots are more similar for mammalian species when removing species of Rodentia and Chiroptera (Figures 1a and 4a). Schoener's D and Hellinger's distance also support this result (Table 1), as FD and SR are more similar when analyzing three groups separately. Considering the fact that Rodentia and Chiroptera comprise more than half of the mammalian species, this result implies that the spatial pattern of SR is greatly impacted by Rodentia and Chiroptera. Therefore, there are risks if we make conservation decisions based on the SR of all mammalian species, as such conservation plans may put too much effect on Rodentia and Chiroptera.

## 5. Conclusions

Based on the comprehensive data of Chinese terrestrial vertebrates, we reconstructed the biodiversity patterns and identified FD and SR hotspots. Most hotspots are in the mountainous regions of southern China, especially southwestern China. Southwestern China harbors FD and SR hotspots of different taxa, which may be due to both environmental and historical factors. FD and SR hotspots are different in each taxon. Mismatches are found in mountains in the north, which have fewer but highly distinct species. Functional hotspots are different among taxa. We also found diversity patterns that differed between distinct groups within a taxonomic unit, such as Anura and Caudata in Amphibia. The FD patterns of Rodentia, Chiroptera and other mammalian species are also different. These results highlight the importance of considering distinct groups separately in conservation actions.

**Supplementary Materials:** The following supporting information can be downloaded at: https://www.mdpi.com/article/10.3390/d14110987/s1, Figure S1: Geographic patterns of functional diversity. (a): Mammals; (b): Birds; (c): Snakes; (d): Lizards; (e): Anura; (f): Caudata. Silhouettes were downloaded from PhyloPic (www.phylopic.org). The map content approval number is GS(2019)1822; Figure S2: Geographic patterns of species richness. (a): Mammals; (b): Birds; (c): Snakes; (d): Squamata species besides snakes; (e): Anura; (f): Caudata. Silhouettes were downloaded from PhyloPic (www.phylopic.org). The map content approval number is GS(2019)1822; Figure S3: Functional diversity of Chiroptera. (a): Functional richness of Chiroptera; (b): Hotspots of Chiroptera which are significantly higher than null models; (c): Hotspots of Chiroptera identified based on 5% threshold; (d): Functional richness of Chiroptera after removing species of Megaderma; (e): Hotspots of Chiroptera which are significantly higher than null models after removing species of *Megaderma*; (f): Hotspots of Chiroptera identified based on 5% threshold after removing species of *Megaderma*. The map content approval number is GS(2019)1822; Figure S4: Functional hotspots identified based on a 5% threshold. (a): Mammals; (b): Birds; (c): Snakes; (d): Squamata species besides snakes; (e): Anura; (f): Caudata. The red color indicates hotspots. Silhouettes were downloaded from PhyloPic (www.phylopic.org). The map content approval number is GS(2019)1822; Figure S5: Geographic distributions of functional diversity (FD) and species richness (SR) of different groups of mammals. (a): FD of mammalian species except for Rodentia and Chiroptera; (b): FD of Rodentia; (c): FD of Chiroptera; (d): SR of mammalian species except for Rodentia and Chiroptera; (e): SR of Rodentia; (f): SR of Chiroptera. The map content approval number is GS(2019)1822; Table S1: Distribution information of amphibian species not recorded in IUCN dataset; Table S2: Functional traits of Snakes; Table S3: Functional traits of Anura (Amphibia) collected by this study. SVL: snout-vent length; HL: head length; HW: head width; ED: eye diameter; TD: tympanum diameter; HAL: hand length; FL: foot length; TL: tibia length; Table S4: Functional traits of Caudata (Amphibia) collected by this study. TOL: total length; SVL: snout-vent length; HL: head length; HW: head width; ED: eye diameter; TL: tail length; FLL: forelimb length; HLL: hindlimb length.

**Author Contributions:** W.Z. conceived the ideas and designed the methodology; X.S. and N.H. collected the data; X.S. and N.H. analyzed the data and W.Z., X.S. and N.H. wrote the manuscript. All authors have read and agreed to the published version of the manuscript.

**Funding:** This work was supported by the National Natural Science Foundation of China (NSFC) under grant No.32170445 and the Fundamental Research Funds for the Central Universities (lzujbky-2021-48).

**Institutional Review Board Statement:** Not applicable.

**Informed Consent Statement:** Not applicable.

**Data Availability Statement:** All data used in this manuscript are available online in additional supporting information which can be found online in the Supplementary Materials section at the end of the article.

**Conflicts of Interest:** The authors declare no conflict of interest.

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
