# Peer review of "Geographical Patterns in Functional Diversity of Chinese Terrestrial Vertebrates"

_diversity, doi:10.3390/d14110987_

Round 1

Reviewer 1 Report

This manuscript describes how spatial patterns of vertebrate biodiversity vary across China and investigates if measures of species richness and functional diversity are correlated and vary among taxa and clades. I commend the authors for an interesting application of these techniques at such a broad geographic scale. Their application should be interesting to conservation biologists, ecologists, and geographers working on global biodiversity conservation. However, I have two main issues with the manuscript detailed below.  

First, the presentation of the data and the results in the manuscript could be improved and I give a few examples of how to improve it below. The multipanel approach for presenting both FD & SR in Figure 1 & 4 is overly complex. Given testing congruence in hotspots using these two measures was an objective of the manuscript, I suggest putting both metrics on the same panel and using different colors to show each metric and the areas of overlapping hotspots. Despite having two figures (3 & 5) the traits themselves are not interpreted much in the results. Please describe the pattern that these figures is intended to show in the Results to set up the discussion of why it is important. Also, the Appendix tables are better suited as supplemental material given the length and complexity of the material.

Second, the manuscript needs to be reviewed by an editor proficient in English. I found numerous English-language flaws on the first couple pages and these continued throughout the manuscript. A few examples of language corrections needed are below.

1 – abstract & introduction - Word choice with Amounts of studies? Suggest replacing with “Numerous studies”

1 – abstract – Correct sentence structure “We find mountains in southern China harbor most hotspots.” Suggest replacing with “We found that the mountains in southern China harbored the most hotspots.”

2 – introduction – “Meanwhile, amounts of new species were descripted recently, especially in amphibians and reptiles” Suggest replacing with “Meanwhile, numerous new species were described recently, especially in amphibians and reptiles”

I elaborate on these general comments with specific comments below.

Specific comments by page:

3 – Please describe your methods for calculating species richness in more detail. As it currently is written, I’m not sure I could replicate your work.

3 – The Appendix tables are difficult to interpret. Providing a few basic statistics on the number of species overall and in each taxa group either in a table or in text would be helpful to the reader in interpreting your results as well as your method of removing grid cells with <5 species and removing two species of Megaderma.

7 – “In this study, we find the hotspots identified based on SR and FD are different in all taxa.” This statement seems to contradict much of your results as they are presented and Figure 1 & 2. Please consider if there is a better way to present your results to highlight these differences.

8 – The last sentence in your Conclusions leaves me wanting more. Please elaborate. What specific conservation actions can your results inform and how do they do that?

11- Appendix/Table A1, 2, 3 & 4 – I suggest making this Appendix and all 4 of its tables supplemental material. The length is distracting from the analyses in the manuscript. The formatting needs to be corrected to ensure all columns are readable and I would suggest making these spreadsheets if possible.

Figure S1 – is there an error in panel b? Why would functional diversity of birds have such a straight line along a specific latitude?

Figure S5 – The silhouettes are more confusing than helpful for this figure. I suggest removing them. Also, it appears the last two sentences of the figure legend belong to another figure.

Author Response

Two main issues with the manuscript:

  1. First, the presentation of the data and the results in the manuscript could be improved and I give a few examples of how to improve it below. The multipanel approach for presenting both FD & SR in Figure 1 & 4 is overly complex. Given testing congruence in hotspots using these two measures was an objective of the manuscript, I suggest putting both metrics on the same panel and using different colors to show each metric and the areas of overlapping hotspots. Despite having two figures (3 & 5) the traits themselves are not interpreted much in the results. Please describe the pattern that these figures is intended to show in the Results to set up the discussion of why it is important. Also, the Appendix tables are better suited as supplemental material given the length and complexity of the material.

>>>> Thank you for your suggestion. We re-made these figures following your suggestions. We described the traits differences between groups and discussed the potential impacts of these differences on functional diversity patterns. We put the appendix tables in supplemental material following your suggestion.

  1. The manuscript needs to be reviewed by an editor proficient in English. I found numerous English-language flaws on the first couple pages and these continued throughout the manuscript. A few examples of language corrections needed are below.

1 – abstract & introduction - Word choice with Amounts of studies? Suggest replacing with “Numerous studies”

1 – abstract – Correct sentence structure “We find mountains in southern China harbor most hotspots.” Suggest replacing with “We found that the mountains in southern China harbored the most hotspots.”

2 – introduction – “Meanwhile, amounts of new species were descripted recently, especially in amphibians and reptiles” Suggest replacing with “Meanwhile, numerous new species were described recently, especially in amphibians and reptiles”

>>>> Thank you for your suggestion. Sorry for our poor English expression. We sent the revised manuscript to a professional language editing company (American Journal Experts: https://www.aje.cn) and asked them to help us edit the manuscript.

Specific comments by page:

3 – Please describe your methods for calculating species richness in more detail. As it currently is written, I’m not sure I could replicate your work.

>>>> Thanks for your suggestion. We calculated species richness based on the distribution of species. We generated the presence-absence matrix using the R package letsR. The grid resolution of grid cells was set as 50 × 50 km2 and the projection was set as Behrmann equal area projection. If the distribution of a species overlapped with a grid, we considered this species distributed in this grid. As some species were not recorded in the distribution datasets, we collected the distribution points of these species, which were listed in Table S1. For these points data, we also generated the presence-absence matrix using the R package letsR. This function generated a dataframe that each row represents a grid and each row represents a species. Species distributed in a grid were recorded as 1. Finally, we combined the two dataframes and then calculated the species richness by adding the number of each row. We wrote more details of the analysis in the revised manuscript following your suggestion.

4 – The Appendix tables are difficult to interpret. Providing a few basic statistics on the number of species overall and in each taxa group either in a table or in text would be helpful to the reader in interpreting your results as well as your method of removing grid cells with <5 species and removing two species of Megaderma.

>>> Thank you very much for your suggestions. We added the details about the information of species in the revised manuscript.

7 – “In this study, we find the hotspots identified based on SR and FD are different in all taxa.” This statement seems to contradict much of your results as they are presented and Figure 1 & 2. Please consider if there is a better way to present your results to highlight these differences.

>>>> Sorry for our poor expression. What we want to say is we can find differences in hotspots identified based on SR and FD in all taxa. We rephrased this part following your suggestion.

8 – The last sentence in your Conclusions leaves me wanting more. Please elaborate. What specific conservation actions can your results inform and how do they do that?

>>> Thanks for your comments! Our study found the spatial patterns of functional diversity were different between distinct groups within a taxonomic unit. These units, such as mammals, birds or amphibians, were widely used in making conservative decisions. Numerous studies identified diversity hotspots or calculated conservation gains for the whole group, such as mammals or birds. We proposed that distinct groups within the taxonomic unit should be treated separately when making conservative decisions. For example, we suggest that Rodentia and Chiroptera of mammals, three orders of amphibians, and snakes of Squamata should be analyzed separately when reconstructing large-scale patterns of functional diversity.

11- Appendix/Table A1, 2, 3 & 4 – I suggest making this Appendix and all 4 of its tables supplemental material. The length is distracting from the analyses in the manuscript. The formatting needs to be corrected to ensure all columns are readable and I would suggest making these spreadsheets if possible.

>>>> Sorry for our mistakes! We changed following your suggestion.

Figure S1 – is there an error in panel b? Why would functional diversity of birds have such a straight line along a specific latitude?

>>>> Sorry for our mistakes! We marked the panels vertically at first. When we changed to mark them horizontally, we did not change the legend. I am very sorry for the mistake! Panel b is the FRic of snakes. As the distribution maps are made based on the specialist’s suggestions, distribution of some species has straight lines along a specific latitude or longitude, sometimes along borders of administrative division. If these species are distinct from other species, the spatial pattern of FD will have a straight line. We checked the distribution of snake species one by one and found such a pattern is caused by Sinomicrurus kelloggi, a unique species of Elapidae. Following the suggestion of another reviewer, we analyzed data using new functional traits databases published recently, which are more informative. For snakes, we also complemented the traits of body length and added new trait data. The spatial pattern of functional diversity was less impacted by Sinomicrurus kelloggi this time. We calculated FRic with and without Sinomicrurus kelloggi and got similar results.

Figure S5 – The silhouettes are more confusing than helpful for this figure. I suggest removing them. Also, it appears the last two sentences of the figure legend belong to another figure.

>>>> Sorry for our mistakes! We changed following your suggestion.

Reviewer 2 Report

Thank you for your manuscript on geographic patterns in the functional diversity of Chinese terrestrial vertebrates. I have looked at your manuscript and read it with great pleasure. I would have one comment: in the introduction, I would disagree with the statement "Regions with high levels of biodiversity, but are threatened by humans, are identified as biodiversity hotspots." I do not think hotspots are defined by human activities. This can be reworded.

The "Acknowledgements" section should be added or deleted altogether.

Tables A2 - A4 are not readable.

Author Response

Thank you for your manuscript on geographic patterns in the functional diversity of Chinese terrestrial vertebrates. I have looked at your manuscript and read it with great pleasure. I would have one comment: in the introduction, I would disagree with the statement "Regions with high levels of biodiversity, but are threatened by humans, are identified as biodiversity hotspots." I do not think hotspots are defined by human activities. This can be reworded.

>>> Thank you very much for your suggestion! We agree that there are confusions about the concept of ‘biodiversity hotspots’. Myers et al. (2000) defined as regions with exceptional concentrations of endemic species were undergoing exceptional loss of habitat as ‘biodiversity hotspots’. However, numerous studies identified diversity hotspots based on diversity metrics alone. Many thanks for your comments. We rephrased this part to make it clear following your suggestion.

The "Acknowledgements" section should be added or deleted altogether.

>>> Thank you for your suggestion! We changed following your suggestion.

Tables A2 - A4 are not readable.

>>> Sorry for our mistakes! We put these tables in supplemental materials.

Reviewer 3 Report

I have reviewed the manuscript “Geographical patterns in functional diversity of Chinese terrestrial vertebrates”. This is a very interesting paper, suitable for publication in Diversity. However, I have the key comments as follows:

1) I do not think that the distribution range maps from the International Union for Conservation of Nature (IUCN) Red List of Threatened Species are not correct for assessing geographical patterns in functional diversity of terrestrial vertebrates in China. The authors should add explanation details to our manuscript. The downloading date should be added to the manuscript.

2) The functional traits should be used for this study based on the following studies:

Yuxi Zhong, Chuanwu Chen, Yanping Wang (2022) A dataset on the life-history and ecological traits of Chinese lizards. Biodiversity Science, 30, 22071. DOI: 10.17520/biods.2022071.

Yunfeng Song, Chuanwu Chen, Yanping Wang (2022) A dataset on the life-history and ecological traits of Chinese amphibians. Biodiversity Science, 30, 22053. DOI: 10.17520/biods.2022053.

Chenchen Ding, Dongni Liang, Wenpei Xin, Chunwang Li, Eric I. Ameca, Zhigang Jiang (2022) A dataset on the morphological, life-history and ecological traits of the mammals in China. Biodiversity Science, 30, 21520. DOI: 10.17520/biods.2021520.

3) How the authors determine the hotspots of functional and species diversities in China? The methods should be detailed.

4) The Chinese maps should be corrected as the same as the policies of Chinese government.

5) Please quantify the overlaps between the hotspots of functional and species diversities in China. E.g., Schoener's D and Hellinger's distance.

Author Response

  1. I do not think that the distribution range maps from the International Union for Conservation of Nature (IUCN) Red List of Threatened Species are not correct for assessing geographical patterns in functional diversity of terrestrial vertebrates in China. The authors should add explanation details to our manuscript. The downloading date should be added to the manuscript.

>>> Thank you for your comments. We agree that distribution data from the IUCN Red List of Threatened Species is the most comprehensive dataset so far. Our analyses are performed based on data from IUCN Red List. But for amphibians, dozens of new species are described each year. These species are not recorded in the distribution dataset of IUCN. Therefore, we collected distributing information on these species (Table S1) and generated the presence-absence matrix combing data from IUCN. The distribution data were downloaded in July 2019. We added this information in the manuscript following your suggestion.

2) The functional traits should be used for this study based on the following studies:

Yuxi Zhong, Chuanwu Chen, Yanping Wang (2022) A dataset on the life-history and ecological traits of Chinese lizards. Biodiversity Science, 30, 22071. DOI: 10.17520/biods.2022071.

Yunfeng Song, Chuanwu Chen, Yanping Wang (2022) A dataset on the life-history and ecological traits of Chinese amphibians. Biodiversity Science, 30, 22053. DOI: 10.17520/biods.2022053.

Chenchen Ding, Dongni Liang, Wenpei Xin, Chunwang Li, Eric I. Ameca, Zhigang Jiang (2022) A dataset on the morphological, life-history and ecological traits of the mammals in China. Biodiversity Science, 30, 21520. DOI: 10.17520/biods.2021520.

>>> Thank you very much for your suggestions. When we performed the analysis, these datasets had not been published. That’s why we did not use these data sets the first time. These datasets are very informative as they contain lots of continuous measurements. For amphibians, we collected continuous measurements by ourselves and most traits in the dataset of Song et al (2022) were categorical data. We combined these data to reconstruct the diversity patterns of Chinese amphibians.

3) How the authors determine the hotspots of functional and species diversities in China? The methods should be detailed.

>>> Thanks for your suggestions. We added more details in the revised manuscript. We identified the hotspots using two methods. First, the top 5% of grids with the highest diversity levels were selected as hotspots. Second, we tested if the diversity of a grid was significantly high following the approach of Shrestha et al. (2021). We randomly selected 1000 grids as random distribution. Then we tested if the species richness and functional richness are significantly higher than the random distribution. For Caudata, we selected 100 grids as random distribution because of fewer data size of this group. The two approaches yielded similar results. We add more details about the methods in the revised manuscript.

4) The Chinese maps should be corrected as the same as the policies of Chinese government.

>>> Thank you for your suggestion. The map content approval number is GS(2019)1822. We added this information in the manuscript.

5) Please quantify the overlaps between the hotspots of functional and species diversities in China. E.g., Schoener's D and Hellinger's distance.

>>> Thank you for your suggestion. We calculated Schoener's D and Hellinger's distance between species richness and functional diversity following your suggestion. To make the indexes more comparable, we first standardized the raster to make all values sum to one. Then, we calculated Schoener's D and Hellinger's distance between FRic and SR for each group.

Round 2

Reviewer 1 Report

Thank you for your time and effort to address my earlier comments. The clarity of the methods as well as presentation and description of the results are much improved. Overall, the English language issues have been addressed and the manuscript is very readable. I have no further comments on the manuscript.